# Breast and Colorectal Cancer Screening Utilization after Hurricane María and the COVID-19 Pandemic in Puerto Rico

**DOI:** 10.3390/ijerph20196870

**Published:** 2023-10-01

**Authors:** Vivian Colón-López, Héctor M. Contreras-Mora, Cynthia M. Pérez, Hérmilis Berríos-Ortiz, Carola T. Sánchez-Díaz, Orville M. Disdier, Nilda Ríos-Morales, Erick L. Suárez-Pérez

**Affiliations:** 1Cancer Control and Population Sciences Division, University of Puerto Rico Comprehensive Cancer Center, San Juan, PR 00927, USA; hermilis.berrios@upr.edu; 2Department of Biostatistics and Epidemiology, University of Puerto Rico Medical Sciences Campus, San Juan, PR 00936, USA; cynthia.perez1@upr.edu (C.M.P.); erick.suarez@upr.edu (E.L.S.-P.); 3Cancer Epidemiology and Health Outcomes, Rutgers Cancer Institute, New Brunswick, NJ 08901, USA; cts109@cinj.rutgers.edu; 4Puerto Rico Statistics Institute, San Juan, PR 00917, USA; orville.disdier@estadisticas.pr.gov (O.M.D.); nilda.rios@estadisticas.pr (N.R.-M.)

**Keywords:** breast cancer, colorectal cancer, COVID-19 pandemic, cancer screening

## Abstract

Puerto Rico (PR) has faced environmental and public health challenges that could have significantly affected cancer screening access. Using administrative claims data from PR’s Medicaid population, this study assessed trends in colorectal and breast cancer screening from 2016 to 2021, the impact of disasters in screening, and the absolute deficit in screening due to the pandemic. The monthly rates of claims were analyzed using Poisson regression. Significant reductions in breast and colorectal cancer screening utilization were observed. The colorectal cancer screening rate in 2017 was 77% lower a month after Hurricanes Irma and María [RR_adj_: 0.23; 95% CI: 0.20, 0.25] compared to the same time period in 2016. Breast cancer screening dropped 50% in November 2017 compared to November 2016 [RR_adj_: 0.50; 95% CI: 0.47, 0.54]. Prospectively, a recovery in utilization has been observed only for breast cancer screening. The results revealed that cancer screening utilization substantially declined after environmental disasters and the pandemic. These findings have potentially severe long-term implications for cancer health disparities and mortality in PR.

## 1. Introduction

In the last five years, Puerto Rico (PR) has endured three major environmental and public health crises (Hurricanes Irma and María, the unprecedented seismic activity in January 2020, and the coronavirus disease of 2019 (COVID-19). After Hurricanes Irma and María struck Puerto Rico, the island remained without electricity for 181 days [1], causing severe damage to Puerto Rico’s infrastructure and the electrical grid. The cumulative impact of these natural disasters added to the pandemic has had profound implications for the health system concerning the provision of oncology care, altering, delaying, or postponing the course of treatment and altering survival outcomes [2]. 

All these disasters undoubtedly have had a detrimental impact early in the cancer continuum spectrum. Cancer prevention and screening in the wake of natural and man-made disasters will exacerbate cancer health disparities in the future [3,4]. The COVID-19 pandemic has significantly hampered the cancer screening infrastructure, services, and programs in many countries [5], affecting treatment adherence [6]. In the United States (US), a recent study that analyzed medical claims using electronic health records showed an abrupt drop—between 86% and 94%—in cancer screening [7], presumably due to access disruptions caused by the COVID-19 national emergency declaration in March 2020. Another study [8] reported declines in screening of up to 90.8% for breast cancer, 79.3% for colorectal cancer, and 63.4% for prostate cancer, with a nearly complete return to pre-COVID-19 monthly screening rates by July for breast and prostate cancers. 

Cancer is the second leading cause of mortality in PR [9,10]. In 2018, breast cancer represented the leading cancer among women in PR, with a mortality rate of 24.5 per 100,000 [10]. Furthermore, colorectal cancer ranks as the second leading cause of cancer incidence and mortality for both men and women, with mortality rates of 26.4 per 100,000 and 17.5 per 100,000 for men and women, respectively [10]. The stage at which colorectal cancer is diagnosed is a critical prognostic factor. Early diagnosis often leads to better treatment outcomes and increased survival rates [11]. However, evidence suggests that patients from the PR Medicaid Program are often diagnosed at more advanced stages [12].

Despite the significant disparities observed for these types of cancers in PR, limited information about breast and colorectal cancer screening utilization is available after Hurricanes Irma and María and the COVID-19 pandemic. In this study, we aim to (1) assess trends in colorectal and breast cancer screening in the Medicaid population of PR from 2016 to 2021, (2) measure the impact of hurricanes and the COVID-19 pandemic on colorectal and breast cancer screening, and (3) estimate the absolute deficit in colorectal and breast cancer screening during (or after) hurricanes and the COVID-19 pandemic.

## 2. Materials and Methods

### 2.1. Data Source and Population

Administrative data claims were requested and obtained from the PR Health Insurance Administration (ASES, by its Spanish acronym) by the PR Statistics Institute [13]. The ASES was established by PR Health Insurance Administration Act No. 72 of 7 September 1993, as amended, to administer the PR Medicaid Program, currently known as Plan Vital, through the establishment of contracts with private insurance providers. As of 1 January 2023, Plan Vital covered approximately 1,297,787 (40%) of the PR population [14]. The effort of this academic-governmental collaboration, which aimed to systematically assess the utilization of health services on the island for the promotion and targeted interventions, has been named PR-TREND. 

We selected two cohorts of enrollees from the database obtained to evaluate monthly and yearly screening rates for breast (women, 50–79 years old) and colorectal cancer (women and men, 40–79 years old). For each year (2016–2021) and month (January–December) studied, we identified female enrollees who met the age requirement for breast cancer screening, according to the US Preventive Services Task Force (USPSTF) [15]. For colorectal cancer screening, in addition to following the USPSTF criteria regarding age [16], we included the 40–49 years cohort in our analysis to comply with local Executive Order Num. 334 of 2015 [17], which order establishes the starting age of the fecal blood occult test to be 40 years of age, due to the high incidence of early-stage colorectal cancer in PR [18].

### 2.2. Data Analysis

The study period analyzed was from 2016 to 2021. We used the Current Procedural Terminology (CPT) and Healthcare Common Procedure Coding System (HCPCS) to retrieve the specific code for the type of cancer screening of interest (colorectal or breast, Appendix A). Beneficiaries with at least one screening test were defined as any individual who met one of the age requirements and had at least one claim related to the screening test of interest (as indicated by the CPT and HCPCS codes). In contrast to the study published by Chen and colleagues [6], we could not exclude participants with a history of the studied cancer to enhance cancer screening accuracy. Instead, we used the most current data (1 January 2020) from the PR Cancer Registry to estimate the prevalence of colorectal and breast cancer and evaluate the effect of this inclusion. Our estimates indicate a colorectal cancer prevalence of 0.3% in the 40–74 age group and a breast cancer prevalence of 0.9% in the 50–74 age group. Given that including individuals with an identified cancer history in our analysis would have had a negligible influence on the results, we decided that age would be the only criterion for selection.

The monthly rate of claims per 100,000 enrollees was computed as follows:(1)$$\mathrm{Rate}\;\mathrm{of}\;\mathrm{claims}=100,000\times \frac{Beneficiaries\;with\;at\;least\;one\;screening\;claim\;per\;month}{Total\;number\;of\;enrollees\;per\;month}$$

These rates were computed per month to describe the trends within the year by sex.

Afterward, the following multivariable Poisson regression model [19] was used to estimate the number of individuals with at least one screening test (colorectal, Appendix A and breast, Appendix A) per year, overall and monthly, while controlling for the total Medicaid enrollment in PR and adjusting for age and sex (colorectal screening):(2)μ=P(i)×exp{βo+Year(i)+Age(j)+Sex(k)}
where μ indicates the expected number of individuals with at least one claim per month in each year; *P*(*i*) indicates the total Medicaid enrollment in year “*i*”; *Year*(*i*) indicates the effect of the year “*i*” relative to reference year (2016); *Age*(*j*) indicates the effect of age-group “*j*”; and *Sex*(*k*) indicates the effect of sex “*k*”.

Based on this model, we estimated the relative risk (*RR*) using 95% confidence intervals (CIs) to assess the relative percent change in the number of enrollees with at least one claim, controlling the total Medicaid enrollment:(3)$$RR(i\;vs.\;2016)=\frac{\mu (i)/P(i)}{\mu (2016)/P(2016)}=exp[Year(i)\pm 1.96\times se]$$
where *se* indicates the standard error of the estimated effect of *Year*(*i*). In case *RR* < 1, the expression 1-*RR* indicates the relative rate reduction, using 2016 as the reference year. In case *RR* > 1, the expression *RR*-1 indicates the relative rate increment, using 2016 as the reference year.

Finally, to determine the screening deficit, based on the rate of claims in 2016, we computed, for each year, the expected number of subjects with at least one claim (number of enrollees times the rate of 2016). Afterward, we computed the deficit as the difference between the number of reported claims minus the expected number of claims (based on the Poisson regression model) for each year after 2016. 

## 3. Results

### 3.1. Trends and the Impact of Hurricanes and COVID-19 on Colorectal Screening in the Medicaid Population of PR from 2016 to 2021

A total of 105,175 beneficiaries of the Medicaid population, 37.7% being men and 62.3% women, had at least one colorectal cancer screening modality claim registered from 2016 through 2021. Overall, a decreasing trend was observed in colorectal cancer screening (Figure 1). Figure 2 shows the rate of colorectal cancer screening, stratified by sex, and Figure 3 by age in relation to environmental disasters and the COVID-19 pandemic. Table 1 shows the RR for each year and each month. Using 2016 as the reference, the RRs (crude and adjusted) of colorectal cancer screening were significantly lower for all years. In 2017 (the year of Hurricanes Irma and Maria), the rate of colorectal cancer screening was 25% lower (RR_adj_: 0.75; 95% CI: 0.74, 0.77) than in 2016. The rate of colorectal cancer screening further dropped in 2020 (earthquakes and the COVID-19 pandemic); in that year, the rate of colorectal cancer screening was 39% lower (RR_adj_: 0.61; 95% CI: 0.60, 0.63) than it had been in 2016. A significantly reduced screening rate (*p* < 0.05) was still observed in 2021, compared with that of 2016 (RR_adj_: 0.73; 95% CI: 0.71, 0.74).

When we evaluated the monthly colorectal cancer screening rate at different periods, a significant reduction (*p* < 0.05) in the rate was observed in all the months of the period of 2017 through 2021 (Table 1). No significant increment (*p* > 0.05) in the colorectal cancer screening rate was observed in any of the years studied compared to 2016. The rate of colorectal cancer screening decreased by 65% (RR_adj_: 0.35; 95% CI: 0.31, 0.38) in September 2017 (Hurricanes Irma and Maria) compared with September 2016. This decrease further dropped in October, with a 77% decrease (RR_adj_: 0.23; 95% CI: 0.20, 0.25) in colorectal cancer screening compared with October 2016. For the following months (November and December 2017), decreases in colorectal cancer screening of 40% (RR_adj_: 0.60; 95% CI: 0.55, 0.65) and 35% (RR_adj_: 0.65; 95% CI: 0.59, 0.71), respectively, were observed when compared to November and December of the reference year. For the months early in the pandemic, a significantly reduced colorectal cancer screening rate was observed in March (15% [RR_adj_: 0.85; 95% CI: 0.83, 0.86]), achieving the highest drop (44% [RR_adj_: 0.56; 95% CI: 0.55, 0.58]) during April, in comparison with April 2016.

### 3.2. Absolute Deficit in Colorectal Cancer Screening

Using the rate of claims for colorectal cancer screening reported in 2016 (Table 2) as a reference, we estimated that the annual deficit in the number of patients with at least one claim was −5360 (95% CI: −5650, −5074) for 2017. During the study period, this deficit shrank after 2017, but in 2020, it reached its negative peak, about −8180 (95% CI: −8462, −7896).

### 3.3. Trends and the Impact of Hurricanes and COVID-19 on Breast Cancer Screening in the Medicaid Population of Puerto Rico from 2016 through 2021

A total of 208,772 women beneficiaries in the Medicaid population had one or more documented breast cancer screening from 2016 through 2021. A variable breast cancer screening uptake rate was observed (Figure 4). Figure 5 shows the rate of breast cancer screening, stratified by age group, during the environmental disasters that occurred during the study period and the COVID-19 pandemic.

A significant increase in breast cancer screening utilization was observed when evaluating the breast cancer screening rates per month and by year (Table 3). This increase in breast cancer screening was observed in all the months from 2017 through 2021, except those months after the disasters. In comparison with the same months in 2016, in 2017, significant declines in breast cancer screening utilization were observed, particularly in the months of October, with a sharp decrease being observed (80% [RR_adj_: 0.20; 95% CI: 0.19, 0.22]), November, with a 50% decrease (RR_adj_: 0.50; 95% CI: 0.47, 0.54), and lastly in December 2017, with a 24% decrease (RR_adj_: 0.76; 95% CI: 0.71, 0.80). There were significant declines in breast cancer screening rates during 2020 compared to the corresponding months in 2016. Specifically, the months of March, April, May, and June witnessed reductions in screening of 12% (RR_adj_: 0.88; 95% CI: 0.86, 0.89), 53% (RR_adj_: 0.47; 95% CI: 0.45, 0.49), 26% (RR_adj_: 0.74; 95% CI: 0.72, 0.75), and 11% (RR_adj_: 0.89; 95% CI: 0.88, 0.90), respectively.

### 3.4. Absolute Deficit in Breast Cancer Screening

Using the rate of claims for breast cancer screening reported in 2016 as a reference, we estimated that the deficit in the number of patients with at least one claim (Table 4) was −4937 (95% CI: −5303, −4572) for 2017 and −6584 (95% CI: −6944, −6223) for 2020. For 2018, 2019, and 2021, increases were observed in the number of patients with at least one claim: 2364 (95% CI: 1991, 2738), 3712 (95% CI: 3353, 4070), and 3632 (95% CI: 3248, 4017), respectively.

## 4. Discussion

To our knowledge, this is the first study to analyze claims data to evaluate breast and colorectal cancer screening in PR after a series of environmental disasters and the COVID-19 pandemic. We examined millions of administrative claims, representing 1.2 million individuals who received Medicaid benefits in PR each year of the period of interest (2016–2021). In this population, over 400,000 men and women were eligible for colorectal cancer screening, and more than 146,000 women were eligible for breast cancer screening. Our study shows significant colorectal and breast screening declines during October, November, and December 2017 (Hurricanes Irma and Maria) and from March through May 2020 (the first three months of the COVID-19 governmental regulations “Lockdown” in PR). By September 2020, the breast cancer screening rates were like those seen during 2016 (reference) and had increased by 2021. The declines in colorectal and breast cancer screening rates during the COVID-19 pandemic follow a pattern quite similar to that observed in Chen and colleagues’ analysis with 60 million people in Medicare Advantage and commercial health plans from across geographically diverse regions of the US from January through July of 2018, 2019, and 2020 [8]. Specifically, similarities with Chen’s study were observed in the sudden drop (in screening) from March through April 2020. The increase in breast cancer screening rate observed in 2021 could be attributed to the implementation of educational campaigns conducted by community-based organizations and other groups during breast cancer awareness and early detection month. For colorectal cancer screening, a partial recovery was seen after July 2020, reaching similar rates as those observed during July 2019. 

Different from the findings of Chen and colleagues [8], as of 2021, our observed rates of colorectal cancer screening for men and women in our study still had not reached the rates observed in 2016, and 2021 had the second lowest yearly rate of all the years analyzed. Similar to our findings, in another study that used the Cosmos database, the authors found that colon cancer screening rates remained slightly below historical baselines, down to 3.4% in 2022, two years into the pandemic [20]. However, these findings are based on a representative sample of patients across all races, sexes, ages, rural/urban locations, and private and public healthcare coverage types. These observed disparities in colorectal cancer screening in the context of PR may be attributed to individual and system-level barriers. Prior studies conducted in PR have highlighted individual barriers to colorectal cancer screening, such as embarrassment, diminished perceived benefits, prevailing fatalism, transportation difficulties (especially in rural areas), limited time, and economic burdens [21]. These individual barriers might have been exacerbated during Hurricanes Irma and Maria, the seismic activity of 2020, and the COVID-19 pandemic, which might reflect these delays in colorectal cancer screening catching up to the 2016 baseline screening rates. 

Nevertheless, the prevalence of these screening tests appears to be low, even in the reference year. In 2016, the average monthly colorectal and breast cancer screening rates were 455 and 2228 per 100,000 enrollees, respectively. An examination of Chen et al.’s 2019 study on a Medicare population revealed an average screening rate of 2262 per 100,000 for colorectal cancer and 4133 per 100,000 for breast cancer in that population [8]. These rates represent approximately five-fold and two-fold increases in colorectal and breast cancer screening rates, respectively, compared to the reference year rates observed in our study. The observed discrepancies may be partially attributed to the differing methodologies and populations between Chen et al.’s study and our own. In Chen et al.’s study, the members of the selected population were beneficiaries of Medicare Advantage and commercial health plans who had a minimum of two years of continuous enrollment prior to the beginning of the month under investigation.

In contrast, the population under study in our research consisted of Medicaid beneficiaries, and the selection was conducted annually. Lastly, the Centers for Medicare and Medicaid Services (CMS) impose more stringent guidelines on the application of HCPCS/CPT codes in the public system, particularly for colorectal cancer screening using fecal occult blood test, immunoassay, and 1–3 simultaneous (iFOBT/FIT) tests. CMS differentiates using the iFOBT/FIT test as a preventative screening measure in asymptomatic individuals, for which the HCPCS billing code G0328 is utilized. Conversely, when individuals visit their primary care facilities with symptoms, discomfort, or other complaints related to gastrointestinal conditions, the iFOBT/FIT test is assigned the CPT billing code 82274 [22]. It is important to note that our analysis excluded these individuals as our study focuses on preventive care. 

## 5. Limitations

Our study has several notable limitations to consider when interpreting the findings. First, our analysis predominantly focuses on individuals insured by Medicaid in PR, given that 1.2 million individuals in PR (or 40% of the total population) are insured by Medicaid, according to the PR Department of Health. This could introduce a bias in our population-level estimates of the cancer screening deficits after the recent environmental disasters and the COVID-19 pandemic since our analysis does not include individuals with private insurance on the island. The Puerto Rico Health Insurance Administration also archives their data every 5–6 years. At the time of our study and data request, the oldest data accessible was from 2016—a unique year as it represents a period without major environmental or public health disasters either in that year or the preceding ones. This limitation in accessing datasets before 2016 restricted our opportunities to access more extended periods of historical data in comparison to 2017 Hurricane Maria, for example. Finally, our adherence to an annual analysis model might not fully align with the USPSTF guidelines, which recommend biennial mammography screenings for women aged 50–74 years, potentially leading to underestimating the observed breast cancer screening rates.

## 6. Conclusions

There is a pressing need for public health initiatives to address the significant deficit in colorectal cancer screening observed within the Puerto Rico Medicaid population. This deficit can be attributed to specific environmental disasters, such as Hurricanes Irma and Maria, earthquakes, and the recent public health crisis brought about by the COVID-19 pandemic. Future research should characterize the array of screening methods, including colonoscopy, iFOBT/FIT, stool DNA tests, and flexible sigmoidoscopy. Also, additional studies should explore how these observed deficits might be explained to other system or contextual factors not examined in this analysis. These challenges, combined with individual barriers, likely obstruct screening rates despite natural disasters. These studies could shed light on the diminishing screening trend evident in the PR Medicaid demographics and prospectively target population-based strategies and monitor the impact of those strategies on cancer prevention. Early preventive screening can pave the way for timely colorectal cancer diagnosis, ultimately leading to more favorable treatment outcomes.

## Figures and Tables

**Figure 1 ijerph-20-06870-f001:**
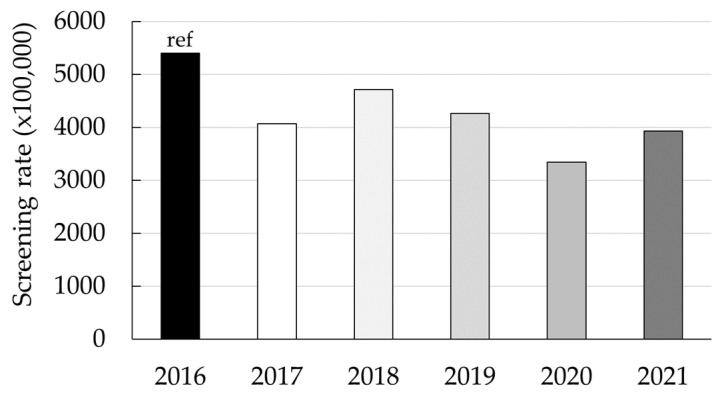
Annual colorectal cancer screening rate, 2016–2021.

**Figure 2 ijerph-20-06870-f002:**
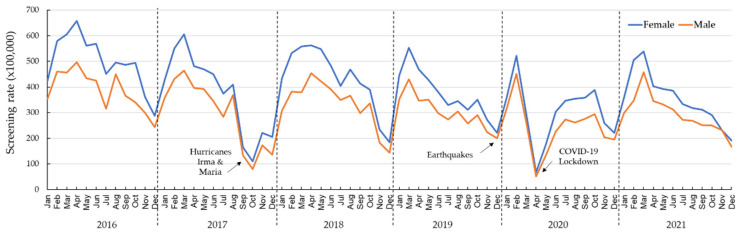
Monthly colorectal cancer screening rates, stratified by sex, of Medicaid beneficiaries, 2016–2021.

**Figure 3 ijerph-20-06870-f003:**
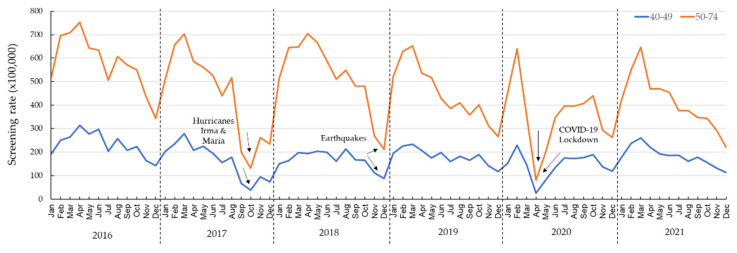
Monthly colorectal cancer screening rates, stratified by age, of Medicaid beneficiaries, 2016–2021.

**Figure 4 ijerph-20-06870-f004:**
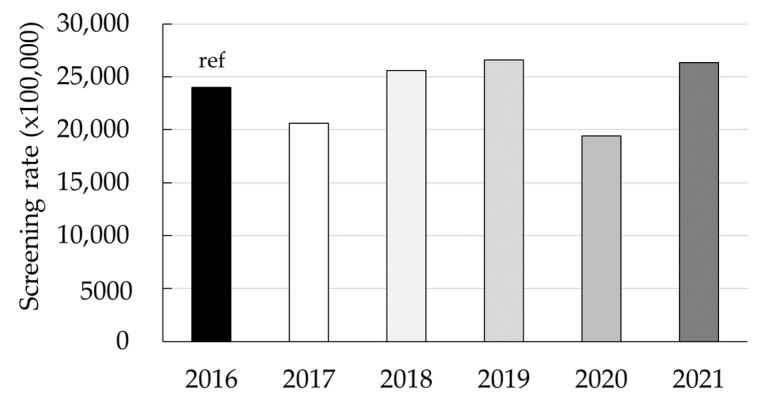
Annual breast cancer screening rate, 2016–2021.

**Figure 5 ijerph-20-06870-f005:**
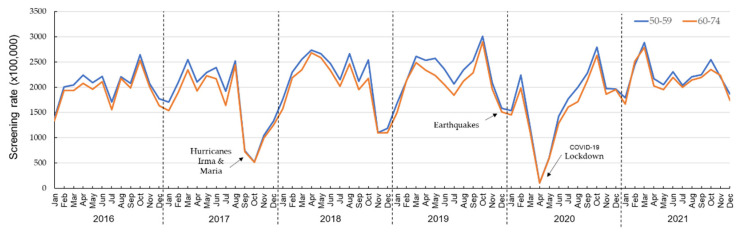
Monthly breast cancer screening rate, stratified by age, of Medicaid beneficiaries, 2016–2021.

**Table 1 ijerph-20-06870-t001:** Colorectal cancer screening rates, 2016–2021.

Year	Number of Beneficiaries (≥1 Screening)	Total Medicaid Enrollment	Rate (×100,000)	RR_crude_ (95% CI)	RR_adj_ ^#^ (95% CI)
Month: Cumulative					
2016	22,046	408,396	5398	1.00	1.00
2017	16,483	404,681	4073	0.75 (0.74, 0.77)	0.75 (0.74, 0.77) *
2018	19,582	415,247	4716	0.87 (0.86, 0.89)	0.87 (0.85, 0.89) *
2019	16,731	392,204	4266	0.79 (0.77, 0.81)	0.78 (0.77, 0.80) *
2020	13,289	397,698	3341	0.62 (0.61, 0.63)	0.61 (0.60, 0.63) *
2021	17,044	433,272	3934	0.73 (0.71, 0.74)	0.73 (0.71, 0.74) *
Month: January					
2016	1638	415,928	394	1.00	1.00
2017	1605	403,358	398	1.01 (0.94, 1.08)	1.00 (0.94, 1.07)
2018	1558	410,884	379	0.98 (0.95, 1.02)	0.98 (0.94, 1.01)
2019	1624	401,217	405	1.01 (0.99, 1.03)	1.00 (0.98, 1.03)
2020	1315	384,998	346	0.97 (0.95, 0.98)	0.96 (0.94, 0.98) *
2021	1376	414,296	332	0.97 (0.95, 0.98)	0.96 (0.95, 0.98) *
Month: February					
2016	2141	405,284	528	1.00	1.00
2017	2008	401,943	500	0.95 (0.89, 1.01)	0.94 (0.89, 1.00)
2018	1936	414,570	467	0.94 (0.91, 0.97)	0.94 (0.91, 0.97) *
2019	1925	399,360	482	0.97 (0.05, 0.99)	0.97 (0.95, 0.99) *
2020	1890	384,527	492	0.98 (0.97, 1.00)	0.98 (0.96, 0.99) *
2021	1857	426,150	436	0.96 (0.95, 0.97)	0.96 (0.95, 0.97) *
Month: March					
2016	2188	404,046	542	1.00	1.00
2017	2219	406,981	545	1.01 (0.95, 1.07)	1.00 (0.95, 1.06)
2018	2012	418,261	481	0.94 (0.91, 0.97)	0.94 (0.91, 0.97) *
2019	1968	393,551	500	0.97 (0.95, 0.99)	0.97 (0.95, 0.99) *
2020	1078	387,514	278	0.85 (0.83, 0.86)	0.84 (0.83, 0.86) *
2021	2161	429,172	504	0.99 (0.97, 1.00)	0.98 (0.97, 1.00)*
Month: April					
2016	2397	407,430	588	1.00	1.00
2017	1806	405,527	445	0.76 (0.71, 0.8)	0.75 (0.71, 0.80) *
2018	2170	421,401	515	0.94 (0.91, 0.96)	0.93 (0.91, 0.96) *
2019	1669	401,605	416	0.89 (0.87, 0.91)	0.89 (0.87, 0.91) *
2020	234	387,408	60	0.57 (0.55, 0.59)	0.56 (0.55, 0.58) *
2021	1631	432,009	378	0.92 (0.90, 0.93)	0.91 (0.90, 0.93) *
Month: May					
2016	2070	408,775	506	1.00	1.00
2017	1765	404,000	437	0.86 (0.81, 0.92)	0.86 (0.81, 0.92) *
2018	2109	426,763	494	0.99 (0.96, 1.02)	0.99 (0.96, 1.02)
2019	1585	402,089	394	0.92 (0.90, 0.94)	0.92 (0.89, 0.94) *
2020	610	391,011	156	0.75 (0.73, 0.76)	0.74 (0.73, 0.76) *
2021	1591	434,186	366	0.94 (0.93, 0.95)	0.94 (0.92, 0.95) *
Month: June					
2016	2069	407,796	507	1.00	1.00
2017	1629	402,434	405	0.80 (0.75, 0.85)	0.79 (0.74, 0.85) *
2018	1885	425,514	443	0.93 (0.91, 0.96)	0.93 (0.91, 0.96) *
2019	1349	390,228	346	0.88 (0.86, 0.90)	0.88 (0.86, 0.90) *
2020	1068	395,432	270	0.85 (0.84, 0.87)	0.85 (0.84, 0.87) *
2021	1534	433,522	354	0.93 (0.92, 0.94)	0.93 (0.92, 0.94) *
Month: July					
2016	1599	406,758	393	1.00	1.00
2017	1327	395,051	336	0.85 (0.79, 0.92)	0.85 (0.79, 0.91) *
2018	1598	419,900	381	0.98 (0.95, 1.02)	0.94 (0.91, 0.97)
2019	1187	388,458	306	0.92 (0.90, 0.94)	0.92 (0.89, 0.94) *
2020	1257	398,514	315	0.95 (0.93, 0.96)	0.94 (0.93, 0.96) *
2021	1336	435,245	307	0.95 (0.94, 0.97)	0.95 (0.94, 0.97) *
Month: August					
2016	1942	407,927	476	1.00	1.00
2017	1563	398,350	392	0.82 (0.77, 0.88)	0.82 (0.77, 0.88) *
2018	1781	419,900	424	0.94 (0.91, 0.97)	0.91 (0.88, 0.95) *
2019	1264	385,079	328	0.88 (0.86, 0.90)	0.88 (0.86, 0.90) *
2020	1263	401,752	314	0.90 (0.89, 0.92)	0.90 (0.88, 0.92) *
2021	1289	435,598	296	0.91 (0.90, 0.92)	0.91 (0.90, 0.92) *
Month: September					
2016	1775	407,794	435	1.00	1.00
2017	607	401,069	151	0.35 (0.32, 0.38)	0.35 (0.31, 0.38) *
2018	1495	410,872	364	0.91 (0.88, 0.95)	0.92 (0.89, 0.95) *
2019	1113	385,933	288	0.87 (0.85, 0.89)	0.87 (0.85, 0.89) *
2020	1309	405,826	323	0.93 (0.91, 0.94)	0.93 (0.91, 0.94) *
2021	1248	437,901	285	0.92 (0.91, 0.93)	0.92 (0.91, 0.93) *
Month: October					
2016	1755	409,855	428	1.00	1.00
2017	390	401,069	97	0.23 (0.20, 0.25)	0.23 (0.20, 0.25) *
2018	1515	413,153	367	0.93 (0.89, 0.96)	0.79 (0.76, 0.83) *
2019	1258	387,035	325	0.91 (0.89, 0.93)	0.91 (0.89, 0.93) *
2020	1423	409,073	348	0.95 (0.93, 0.97)	0.95 (0.93, 0.96) *
2021	1201	439,472	273	0.91 (0.90, 0.93)	0.91 (0.90, 0.93) *
Month: November					
2016	1371	409,581	335	1.00	1.00
2017	792	395,051	200	0.60 (0.55, 0.65)	0.60 (0.55, 0.65) *
2018	859	404,876	212	0.80 (0.76, 0.83)	0.79 (0.76, 0.83) *
2019	969	385,797	251	0.91 (0.88, 0.93)	0.91 (0.88, 0.93) *
2020	970	412,034	235	0.92 (0.90, 0.93)	0.91 (0.90, 0.93) *
2021	1022	440,377	232	0.93 (0.91, 0.94)	0.93 (0.91, 0.94) *
Month: December					
2016	1101	409,581	269	1.00	1.00
2017	772	441,340	175	0.65 (0.59, 0.71)	0.65 (0.59, 0.71) *
2018	664	396,867	167	0.79 (0.75, 0.83)	0.78 (0.75, 0.82) *
2019	820	386,100	212	0.92 (0.90, 0.95)	0.92 (0.89, 0.95) *
2020	872	414,283	210	0.94 (0.92, 0.96)	0.94 (0.92, 0.96) *
2021	798	441,340	181	0.92 (0.91, 0.94)	0.92 (0.91, 0.94) *

^#^ Adjusted for age and sex. * *p* < 0.05. Note: Significant interaction terms (*p* < 0.05) for age and month were shown in the Poisson regression model. Therefore, the year comparisons were stratified by month. Also, the results showed additional significant (*p* < 0.05) interaction terms assessed with the likelihood-ratio test; however, no further stratification was performed due to the limited sample size.

**Table 2 ijerph-20-06870-t002:** Colorectal cancer screening deficit per year: 2017–2021.

Year	Number of Beneficiaries Observed (≥1 Screening Claim)	Total Medicaid Enrollment	Number of Beneficiaries Expected (≥1 Screening Claim) *	Annual Deficit (95% CI)
2016	22,046	408,396	-	-
2017	16,483	404,681	21,845	−5360 (−5650, −5074)
2018	19,582	415,247	22,416	−2835 (−3129, −2537)
2019	16,731	392,204	21,172	−4440 (−4720, −4161)
2020	13,289	397,698	21,468	−8180 (−8462, −7896)
2021	17,044	433,272	23,389	−6345 (−6653, −6036)

* Using the annual rate of claims from 2016.

**Table 3 ijerph-20-06870-t003:** Breast cancer screening rates, 2016–2021.

Year	Number of Beneficiaries (≥1 Screening)	Total Medicaid Enrollment	Rate (×100,000)	RR_crude_ (95% CI)	RR_adj_ ^#^ (95% CI)
Month: Cumulative					
2016	34,914	145,541	23,989	1.00	1.00
2017	29,903	145,234	20,590	0.86 (0.85, 0.87)	0.86 (0.85, 0.87)
2018	37,948	148,332	25,583	1.07 (1.05, 1.08)	1.07 (1.05, 1.08)
2019	37,916	142,582	26,592	1.11 (1.09, 1.12)	1.11 (1.10, 1.13)
2020	27,781	143,251	19,393	0.81 (0.80, 0.82)	0.81 (0.80, 0.82)
2021	40,310	152,893	26,365	1.10 (1.08, 1.10)	1.10 (1.09, 1.12)
Month: January					
2016	2017	147,013	1372	1.00	1.00
2017	2370	144,587	1639	1.19 (1.13, 1.27)	1.20 (1.13, 1.27) *
2018	2471	146,947	1682	1.11 (1.08, 1.14)	1.11 (1.08, 1.14) *
2019	2338	145,619	1606	1.05 (1.03, 1.07)	1.06 (1.03, 1.08) *
2020	2106	140,161	1503	1.02 (1.01, 1.04)	1.02 (1.01, 1.04) *
2021	2571	147,739	1742	1.05 (1.04, 1.06)	1.05 (1.04, 1.06) *
Month: February					
2016	2859	144,366	1980	1.00	1.00
2017	2911	144,111	2020	1.02 (0.97, 1.07)	1.02 (0.97, 1.07)
2018	3332	147,877	2253	1.07 (1.04, 1.09)	1.07 (1.04, 1.09) *
2019	3086	144,633	2134	1.03 (1.01, 1.04)	1.03 (1.01, 1.04) *
2020	2982	139,864	2132	1.02 (1.01, 1.03)	1.02 (1.01, 1.03) *
2021	3752	151,589	2475	1.05 (1.04, 1.06)	1.05 (1.04, 1.06) *
Month: March					
2016	2883	144,025	2002	1.00	1.00
2017	3597	145,799	2467	1.23 (1.17, 1.29)	1.23 (1.17, 1.30) *
2018	3672	148,920	2466	1.11 (1.08, 1.14)	1.11 (1.08, 1.14) *
2019	3642	142,302	2559	1.09 (1.07, 1.10)	1.09 (1.07, 1.10) *
2020	1651	140,651	1174	0.88 (0.86, 0.89)	0.88 (0.86, 0.89) *
2021	4336	152,488	2844	1.07 (1.06, 1.08)	1.07 (1.06, 1.08) *
Month: April					
2016	3160	145,287	2175	1.00	1.00
2017	2961	145,477	2035	0.94 (0.89, 0.98)	0.94 (0.89, 0.98) *
2018	4070	149,846	2716	1.12 (1.09, 1.14)	1.12 (1.09, 1.14) *
2019	3544	144,700	2449	1.04 (1.02, 1.06)	1.04 (1.03, 1.06) *
2020	152	140,516	108	0.47 (0.45, 0.49)	0.47 (0.45, 0.49) *
2021	3224	152,896	2108	0.99 (0.98, 1.00)	0.99 (0.98, 1.00)
Month: May					
2016	2974	145,886	2039	1.00	1.00
2017	3288	145,084	2266	1.11 (1.06, 1.17)	1.11 (1.06, 1.17) *
2018	3984	151,411	2631	1.14 (1.11, 1.16)	1.14 (1.11, 1.16) *
2019	3538	145,890	2425	1.06 (1.04, 1.08)	1.06 (1.05, 1.08) *
2020	852	141,509	602	0.74 (0.72, 0.75)	0.74 (0.72, 0.75) *
2021	3085	153,396	2011	1.00 (0.99, 1.01)	1.00 (0.99, 1.01)
Month: June					
2016	3155	145,202	2173	1.00	1.00
2017	3326	144,560	2301	1.06 (1.01, 1.11)	1.06 (1.01, 1.11) *
2018	3621	150,206	2412	1.05 (0.03, 1.08)	1.05 (1.03, 1.08) *
2019	3173	142,527	2226	1.01 (0.99, 1.02)	1.01 (0.99, 1.03)
2020	1950	142,643	1367	0.89 (0.88, 0.90)	0.89 (0.88, 0.90) *
2021	3451	152,808	2258	1.01 (1.00, 1.02)	1.01 (1.00, 1.02)
Month: July					
2016	2391	145,030	1649	1.00	1.00
2017	2577	142,575	1807	1.10 (1.04, 1.16)	1.10 (1.04, 1.16) *
2018	3123	148,888	2098	1.13 (1.10, 1.16)	1.13 (1.10, 1.16) *
2019	2801	142,176	1970	1.06 (1.04, 1.08)	1.06 (1.04, 1.08) *
2020	2442	143,339	1704	1.01 (0.99, 1.02)	1.01 (1.00, 1.02)
2021	3101	153,277	2023	1.04 (1.03, 1.05)	1.04 (1.03, 1.05) *
Month: August					
2016	3195	145,477	2196	1.00	1.00
2017	3578	143,861	2487	1.13 (1.08, 1.19)	1.13 (1.08, 1.19) *
2018	3845	148,888	2582	1.08 (1.06, 1.11)	1.08 (1.06, 1.11) *
2019	3162	140,523	2250	1.01 (0.99, 1.02)	1.01 (0.99, 1.03)
2020	2709	144,128	1880	0.96 (0.95, 0.97)	0.96 (0.95, 0.98) *
2021	3345	153,315	2182	1.00 (0.99, 1.01)	1.00 (0.99, 1.01)
Month: September					
2016	2972	145,500	2043	1.00	1.00
2017	1069	144,837	738	0.36 (0.34, 0.39)	0.36 (0.34, 0.39) *
2018	3001	146,218	2052	1.00 (0.98, 1.03)	1.00 (0.98, 1.03)
2019	3410	140,621	2425	1.06 (1.04, 1.08)	1.06 (1.04, 1.08) *
2020	3222	145,262	2218	1.02 (1.01, 1.03)	1.02 (1.01, 1.03) *
2021	3428	153,943	2227	1.02 (1.01, 1.03)	1.02 (1.01, 1.03) *
Month: October					
2016	3800	146,162	2599	1.00	1.00
2017	754	144,837	521	0.20 (0.19, 0.22)	0.20 (0.19, 0.22) *
2018	3551	148,924	2384	0.96 (0.94, 0.98)	0.96 (0.94, 0.98) *
2019	4170	140,881	2960	1.04 (1.03, 1.06)	1.05 (1.03, 1.06) *
2020	3978	146,124	2722	1.01 (1.00, 1.02)	1.01 (1.00, 1.02) *
2021	3800	154,344	2462	0.99 (0.98, 1.00)	0.99 (0.98, 1.00) *
Month: November					
2016	2993	146,269	2046	1.00	1.00
2017	1468	142,575	1029	0.50 (0.47, 0.54)	0.50 (0.47, 0.54) *
2018	1611	146,815	1097	0.73 (0.71, 0.75)	0.73 (0.71, 0.76) *
2019	2863	140,373	2040	1.00 (0.98, 1.02)	1.00 (0.98, 1.02)
2020	2836	147,085	1928	0.99 (0.97, 1.00)	0.99 (0.97, 1.00) *
2021	3418	154,422	2213	1.02 (1.01, 1.03)	1.02 (1.01, 1.03) *
Month: December					
2016	2515	146,269	1719	1.00	1.00
2017	2004	154,499	1297	0.75 (0.71, 0.80)	0.76 (0.71, 0.80) *
2018	1667	145,045	1149	0.82 (0.79, 0.84)	0.82 (0.79, 0.85) *
2019	2189	140,743	1555	0.97 (0.95, 0.99)	0.97 (0.95, 0.99) *
2020	2901	147,733	1964	1.03 (1.02, 1.05)	1.03 (1.02, 1.05) *
2021	2799	154,499	1812	1.01 (1.00, 1.02)	1.01 (1.00, 1.02) *

^#^ Adjusted for age and sex. * *p* < 0.05. Note: Significant interaction terms (*p* < 0.05) for age and month were shown in the Poisson model. Therefore, the year comparisons were stratified by month. Also, the results showed additional significant (*p* < 0.05) interaction terms assessed with the likelihood-ratio test; however, no further stratification was performed due to t he limited sample size.

**Table 4 ijerph-20-06870-t004:** Breast cancer screening deficit.

Year	Number of Beneficiaries Observed (≥1 screening)	Total Medicaid Enrollment	Number of Beneficiaries Expected (≥1 Screening) *	Annual Deficit (95% CI)
2016	34,914	145,541	-	-
2017	29,903	145,234	34,840	−4937 (−5303, −4572)
2018	37,948	148,332	35,584	2364 (1991, 2738)
2019	37,916	142,582	34,204	3712 (3353, 4070)
2020	27,781	143,251	34,365	−6584 (−6944, −6223)
2021	40,310	152,893	36,678	3632 (3248, 4017)

* Using the annual rate of claims from 2016.

## Data Availability

Third Party Data. Data was obtained from the Puerto Rico Health Insurance Administration, a government agency, and is unavailable for sharing.

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
