# Peer review of "Breast and Colorectal Cancer Screening Utilization after Hurricane María and the COVID-19 Pandemic in Puerto Rico"

_ijerph, 2023, doi:10.3390/ijerph20196870_

Round 1

Reviewer 1 Report

Authors have used the administrative claims data from PR's Medicaid population and tried to corelate the environmental and public health challenges with colorectal and breast cancer screening. Below is my comments

1. Lack of evidence to justify the title of manuscript, this study not able to establish any co relation between environmental and public health challenges and colorectal and breast cancer screening. 

2. Why only 2016 have been marked as reference? Why this study focused with individual  insured by Medicaid only?

3. Institutional Review Board Statement: Ethical review and approval were waived for this study  because of its use of deidentified data and there being no human subjects... deidentified data is sufficient to waive off Ethical review??

4. Section 5: Limitation of study..(Our study focused primarily on individuals insured by Medicaid).. why authors trying to co relate the environmental and public health challenges  with colorectal and breast cancer screening only, what about the common other disorders.??

5. Conclusion not representing the study..

6. I think additional scientific evidences required to co-relate the environmental and public health challenges with colorectal and breast cancer screening.

Author Response

Thank you very much for taking the time to review this manuscript. Please see the attached document. We have updated the manuscript to reflect the changes discussed in the responses and it is ready for submission at your earliest convenience.

Reviewer 2 Report

This paper examines a critical public health issue facing not only Puerto Rico, but other communities throughout the world. I found the manuscript very well written, but would benefit from additional information in the introduction and discussion. My edits are described below:

1. Line 40: Please utilize the proper citation format. Mention where this study was conducted. 

2. Introduction: I would recommend adding a bit more information regarding why preventive screenings were hindered during the natural disasters.

3. Can you please explain why 2016 was selected as the reference year?

4. Lines 183-187: The sentence is confusing to understand.

5. Can you include your model parameters in a separate table in the supplementary materials?

5. The citations in the first paragraph of the discussion should be updated to the proper in-text format. I would recommend not referring to the Chen study informally and instead use the proper citation. 

6. Are there other studies you can refer to when evaluating/comparing your findings in the discussion? Additional discussion regarding how to mitigate the reduction in screenings should be included.

7. Your limitations focus on breast cancer screenings. Is there any additional information to add regarding colorectal cancer screenings?

8. The conclusion is only one sentence and a brief summary. Additional thoughts regarding how this data can be used, what should be done, and future work should be added. 

no comments to add

Author Response

(The authors gave the same response as above.)

Reviewer 3 Report

It is up to the authors but not sure if the table with rates for each month are necessary given that the same information is provided graphically. Also, in the methods could the authors provide a clearer explanation as to how the rates were computed.  Perhaps give an example of the computation by putting numbers in the equations.  Could also the authors in the discussion provide some speculation on how the different disasters may have caused the declines such as facilities closed, transportation issues, medicare overwhelmed with higher priorties. 

no comments

Author Response

(The authors gave the same response as above.)

Round 2

Reviewer 1 Report

"The authors have made substantial revisions to the manuscript. However, there are still questions regarding the scope of this study."

1. Methods section Line 100 says.. “The University of PR Comprehensive Cancer Center Institutional Review Board deemed the study exempt from review, and patient informed consent requirements were waived, both because deidentified data were used and because the study did not involve human subjects research (IRB 2022-09-83). " Simultaneously Institutional Review Board Statement: line 444 says “This study was evaluated and approved by expedited review. This study met criteria #4 for exempt studies: Section 46.104(d)(4), secondary research for  which consent is not required.." .. It created additional confusion Kindly review the above sections and do the needful.

2. We would highly appreciate the inclusion of other statistical tools for correlation analysis, such as the Chi-Square Test of Independence and Logistic Regression for additional evidence of co relation. Additionally, the incorporation of in silico statistical tests would be valuable."

Reviewer 2 Report

The revisions are sufficient. 

In the discussion, please make sure that citations are properly included. When referring to Chen et al., you must include the year or the numerical citation as well. The conclusion has some typos and I would recommend reviewing and editing once more (extra words, sentences don't make sense, etc.)
